# Distilling Structural Representations into Protein Sequence Models

**Jeffrey Ouyang-Zhang, Chengyue Gong, Yue Zhao, Philipp Krähenbühl, Adam R. Klivans, Daniel J. Diaz**
University of Texas at Austin
`{jozhang,cygong17,yzhao,philkr,klivans,danny.diaz}@utexas.edu`

## Abstract

Protein language models, like the popular ESM2, are widely used tools for extracting evolution-based protein representations and have achieved significant success on downstream biological tasks. Representations based on sequence and structure models, however, show significant performance differences depending on the downstream task. A major open problem is to obtain representations that best capture both the evolutionary and structural properties of proteins in general. Here we introduce *Implicit Structure Model* (*ISM*), a sequence-only input model with structurally-enriched representations that outperforms state-of-the-art sequence models on several well-studied benchmarks including mutation stability assessment and structure prediction. Our key innovations are a microenvironment-based autoencoder for generating structure tokens and a self-supervised training objective that distills these tokens into ESM2's pre-trained model. We have made *ISM*'s structure-enriched weights easily available: integrating ISM into any application using ESM2 requires changing only a single line of code. Our code is available at https://github.com/jozhang97/ISM.

## 1 Introduction

Protein language models (pLMs) are versatile feature extractors with proven success across numerous downstream applications (Elnaggar et al., 2021; Brandes et al., 2022; Rives et al., 2019; Lin et al., 2022). Their accessibility has significantly democratized protein research, enabling biologists with limited computational expertise to apply advanced machine learning techniques to their specific protein domain. The method's success comes from its exclusive use of sequences, bypassing costly, unreliable, or infeasible structure computations and sophisticated data-engineering pipelines.

The tradeoff is that pLMs are often lack structural context and underperform (relative to structure-based models) on tasks that typically require structural insight (Su et al., 2023; Yang et al., 2023; Zhang et al., 2024; Gaujac et al., 2024; Frolova et al., 2024; Li et al., 2024; Kulikova et al., 2023; Allman et al., 2024). Longstanding biological research (Anfinsen, 1973) does suggest that the amino acid sequence is solely responsible for the folding of the structure. Indeed, sequence-only models trained using masked language modeling learn to extract structure features encoded in evolutionary co-variations (Lin et al., 2022). However, current state-of-the-art frameworks, such as AlphaFold, require the protein's evolutionary history as an additional input, demonstrating that sequence-only models fail to extract all the structural information within a multiple sequence alignment (MSA). Building a *single-sequence* model (without additional MSA input) that leads to structurally-informed representations remains a challenging open problem.

In this paper, we introduce *Implicit Structure Model* (*ISM*), a sequence-only protein language model that is trained to *implicitly* capture structural information. Our key contribution is a new self-supervised pre-training objective, *structure-tuning*, where the sequence model learns to distill features derived from structure-based models (see Figure 1). As a result, *ISM* outperforms sequence-only models and is competitive with pLM frameworks that *explicitly* take the protein structure as an additional input. For example, on the CAMEO protein structure prediction benchmark *ISM* outperforms its ESM2 counterpart with a GDT-TS score of 0.67 versus 0.64 (see Table 1). For S669 $\Delta\Delta G$ prediction, *ISM* surpasses ESM2 in AUC (0.76 vs 0.72) and even matches specialized models

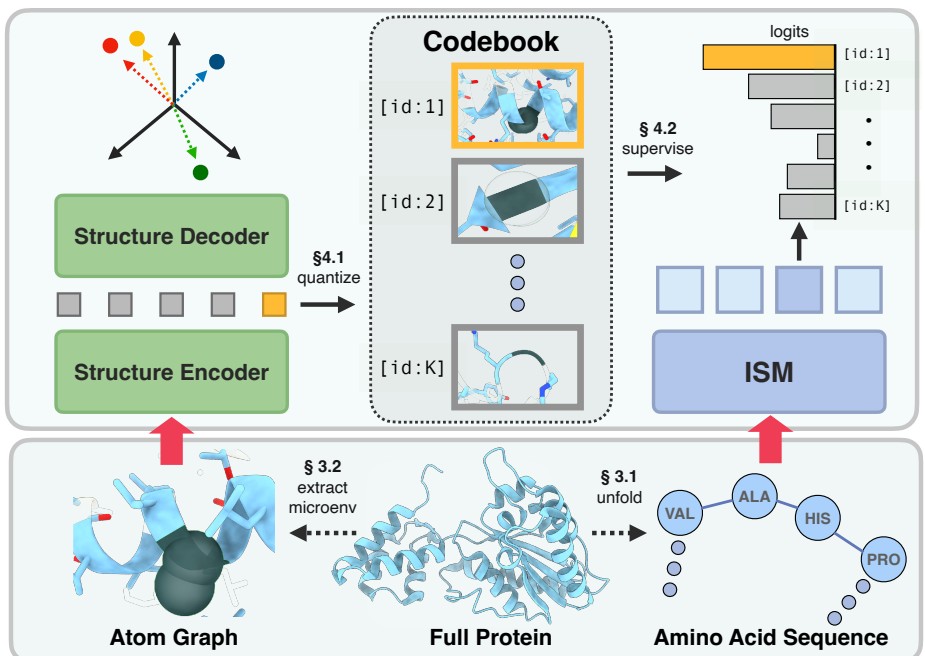

Figure 1: **Structure-tuning a protein language model.** *Implicit Structure Model* (*ISM*) is a sequence-only protein model (right) supervised by structure tokens derived from a structure model (left). For every residue, a structure encoder takes the atoms of a residue's microenvironment as input and produces a structural representation. We discretize these representations into tokens using a codebook extracted via k-means clustering. The *ISM sequence model* learns to predict these structure tokens.

that process atomic environments (0.76 vs 0.75, see Table 2). Our results align with prior works that show multiple modalities enhance model performance (Gong et al., 2024; Hayes et al., 2024).

Structure-tuning is a fine-tuning technique where a sequence-only model is trained to predict structure tokens – rather than masked amino acids – for each protein residue (see Figure 1). Our structure tokens, derived from our Atomic Autoencoder and MutRank (Gong et al., 2024), capture key chemical interactions that underpin the protein's tertiary structure. Structure-tuning distills these structural representations into *ISM*, as demonstrated by the significant improvement in predicting long-range tertiary interactions (0.49 vs 0.35, see Table 1).

## 2 RELATED WORK

**Protein Language Models.** These models take an amino acid sequence as input and produce a deep representation for each amino acid conditioned on the entire sequence. Commonly-used models such as ProtBERT, ProteinBERT, ESM1b, and ESM2 use transformer-based architectures and are trained to maximize wildtype accuracy (*i.e.,* reconstruct masked amino acids) (Elnaggar et al., 2021; Brandes et al., 2022; Rives et al., 2019; Lin et al., 2022).

One of the motivations behind ESM2 was to build a single-sequence variant of AlphaFold that does not require the computationally expensive task of generating MSAs. The resulting model, ESMFold, is a widely used tool but generally underperforms when compared to AlphaFold in terms of predicted structural quality. This demonstrates that ESM2 does not fully capture the epistatic landscape induced during evolution. This has motivated research on augmenting sequence models with a structural modality, and we describe some of these works below.

**Sequence models with structure loss.** The ESM2-s sequence model incorporates structural information by fine-tuning ESM2 to predict a protein's structural fold (Zhang et al., 2024). The fold of a protein, however, is biologically coarse-grained information. *ISM* achieves superior performance by using the more fine-grained approach of training at the residue level. More specifically, in our training objective, each residue is tasked with predicting its corresponding local structural environment.

S-PLM and "Structure-infused protein language models (SIPLM)" use a type of CLIP training to align sequence and structural features (Wang et al., 2023; Peñaherrera & Koes, 2023). This technique is also coarse-grained because its training objective does not operate at a residue level (we do not include SIPLM in our tables of results due to its relatively weak performance on our benchmarks).

AlphaFold also learns structural representations from sequences (Jumper et al., 2021). However, it requires a multiple sequence alignment as input, which is expensive to compute and often unavailable for many practical applications. Furthermore, prior works have shown that Evoformer, the feature extractor for AlphaFold, underperforms ESM2 on various downstream tasks that involve less structural information (Hu et al., 2022). On these tasks, *ISM* still achieves comparable performance to ESM2.

**Sequence models with structure inputs.** These models extend sequence models by using the structure as an additional input. SaProt (Su et al., 2023) and ProstT5 (Heinzinger et al., 2023) use the VQVAE from FoldSeek (van Kempen et al., 2022) to extract per-residue structure tokens as additional inputs to a protein language model. MULAN (Frolova et al., 2024) extends these works to include structural features (torsion angles) as additional inputs. Similarly, ProSST (Li et al., 2024) also takes structural tokens as inputs. However, instead of using FoldSeek tokens, ProSST trains a Denoising Autoencoder to extract per-residue features, which are then tokenized into a structure sequence using K-means clustering. All these models require a protein structure as input at inference time. There are well-known drawbacks to frameworks requiring structure as input. In addition to requiring a more sophisticated data engineering pipeline, there are some cases where the structure has not been experimentally resolved and cannot be accurately modeled using computational tools (*e.g.,* antibody-antigen complexes, conformer specific protein-protein interactions, post-translation modification-dependent conformations, interfaces, etc).

**Protein Structure Autoencoders.** These autoencoders are structure-based models that take the backbone atom coordinates as input and encode each residue into a discrete token (Gaujac et al., 2024; Hayes et al., 2024). The sequence of discrete tokens is used to reconstruct the positions of backbone atoms using coordinate losses (*e.g.,* frame aligned point error, distogram classification). Protein structure denoising Autoencoders take a noisy variant of the protein backbone as input and then learn a latent embedding that decodes the backbone atoms (Peñaherrera & Koes, 2023; Li et al., 2024). Foldseek (van Kempen et al., 2022) extracts features for a residue given the backbone geometry of its nearest neighbors. Unlike our approach, these works use only the protein backbone as input. We also train a structural autoencoder, but instead of reconstructing the local backbone of a protein, we reconstruct the coordinates of all atoms within the local chemical environment surrounding a masked residue (masked microenvironment).

## 3 PRELIMINARIES

Let $\boldsymbol{x}_{\text{seq}} = (x_1, ..., x_L)$ be a protein sequence of $L$ amino acids where each amino acid residue $x_l \in \{A, C, ..., Y\}$. The atoms defined by this sequence fold into an energetically favorable 3-dimensional structure $\boldsymbol{x}_{\text{struct}} = \{(p_i, e_i, \boldsymbol{c}_i)\}_{i=1}^{N}$ where each atom $i$ consists of residue sequence position $p_i \in \{1, ..., L\}$, an element type $e_i \in \{C, H, N, O, P, S, X\}$ and coordinates $\boldsymbol{c}_i \in \mathbb{R}^3$.

### 3.1 PROTEIN SEQUENCE MODELS

A protein language model $\textbf{pLM}$ takes a protein sequence $\boldsymbol{x}_{\text{seq}}$ as input and produces a latent representation $\textbf{pLM}(\boldsymbol{x}_{\text{seq}}) \in \mathbb{R}^{L \times D}$ for downstream tasks. Most models use a transformer architecture and are pre-trained via a masked language modeling (MLM) loss. During training, a subset $\mathbb{M} \subset \{1, ..., L\}$ of the sequence is replaced with the `[mask]` token $\tilde{x}_i = \begin{cases} \texttt{[mask]} & \text{if } i \in \mathbb{M} \\ x_i & \text{otherwise} \end{cases}$ with $\tilde{\boldsymbol{x}}_{\text{seq}} = (\tilde{x}_1, ..., \tilde{x}_L)$. The model learns to reconstruct the masked tokens with

$$\mathcal{L}_{\text{MLM}} = \frac{1}{|\mathbb{M}|} \sum_{i \in \mathbb{M}} \ell_{\text{CE}}(\boldsymbol{C}_{\text{MLM}}^{\top} \textbf{pLM}(\tilde{\boldsymbol{x}}_{\text{seq}})_i, x_i), \qquad (1)$$

for the cross entropy loss $\ell_{\text{CE}}$, indexed feature $\textbf{pLM}(\tilde{\boldsymbol{x}}_{\text{seq}})_i \in \mathbb{R}^D$ at position $i$, and a linear classification head $\boldsymbol{C}_{\text{MLM}}$ that predicts the amino acid type. While the backbone $\textbf{pLM}$ is used for downstream tasks, $\boldsymbol{C}_{\text{MLM}}$ is only used for pre-training.

## 3.2 Protein Structure Models

An all-atom protein structure model **pSM** computes an atom-level feature representation from the local geometric description of each residue. It starts from a microenvironment $x_{\text{microenv}}^l$ that contains all atoms in a radius $r = 10\text{Å}$ around $\boldsymbol{\alpha}_l \in \mathbb{R}^3$, the coordinates of the $\alpha$-carbon of residue $l$:

$$x_{\text{microenv}}^l = \{(e_i, c_i) : \forall i \in \{1, ..., N\} \text{ such that } ||c_i - \boldsymbol{\alpha}_l|| < r\}.$$

A common backbone for protein structure models is a Graph Transformer $G$ (Ying et al., 2021). The graph transformer $G(x_{\text{microenv}}^l)$ embeds each atom's element type $e_i$ in a set $\mathbf{e} = \{e_1, ..., e_{n'}\}$, where $n'$ is the number of atoms in the microenvironment. In attention updates, the graph transformer adds an attention bias $B_{ij}^l = ||c_i - c_j||$ based on the pairwise distance between atoms $i$ and $j$. This attention bias $B^l$ is the only structural information given to the transformer. The graph transformer then produces a set of output features $\{z_1^l, ..., z_{n'}^l\} = G(x_{\text{microenv}}^l)$, one per input atom $e_i$. The graph transformer is commonly trained on the downstream task using a supervised learning objective (Ying et al., 2021). In this work, we use the Graph Transformer directly to train a structure model on atomic reconstructions of proteins in our pre-training dataset.

MutComputeX-GT (Diaz et al., 2024) pre-trains a Graph Transformer using a structural analog of masked language modeling. They define a masked microenvironment $x_{\text{masked-microenv}}^l$ that contains all atoms of other residues $p_i \neq l$

$$x_{\text{masked-microenv}}^l = \{(e_i, c_i) : \forall i \in \{1, ..., N\} \text{ such that } p_i \neq l \text{ and } ||c_i - \boldsymbol{\alpha}_l|| < r\},$$

and pool all-atom level features into a single residue level embedding $z^l = \frac{1}{n} \sum_i z_i^l$ for $\{z_1^l, ..., z_n^l\} = G(x_{\text{masked-microenv}}^l)$ where $n$ is the number of atoms in the masked microenvironment. They then predict the masked-out amino acid type $x_l$:

$$\mathcal{L}_{\text{AA}}^l = \ell_{\text{CE}}(\boldsymbol{C}_{\text{AA}}^\top z^l, x_l). \tag{2}$$

where $\boldsymbol{C}_{\text{AA}}$ is a linear classification head.

MutRank (Gong et al., 2024) uses the EvoRank self-supervised training objective to learn the evolutionary mutational landscape of a residue from the masked microenvironment. More specifically, it learns to predict an evolutionary score derived from the protein's multiple sequence alignment.

## 4 Method

*ISM* is a sequence model that takes as input only an amino acid sequence $x_{\text{seq}} = (x_1, ..., x_L)$ but is trained to implicitly capture structural information. We start by training an Atomic Autoencoder, based on a Graph Transformer, on protein structures. The autoencoder is trained with a geometric reconstruction loss and the MutComputeX-GT objective $\mathcal{L}_{\text{AA}}^l$. We then cluster the resultant features into one of $K$ structure tokens. We use the sequence $\boldsymbol{s} = (s_1, ..., s_L)$ of structure tokens $s_l \in \{1, ..., K\}$ as additional supervisory signal for the sequence-only *Implicit Structure Model* (*ISM*).

### 4.1 Atomic Autoencoder

Atomic Autoencoder uses an encoder-decoder architecture with a Graph Transformer encoder and a plain transformer decoder. The encoder takes the masked microenvironment $x_{\text{masked-microenv}}^l$ as input and produces atomic representations $\{z_1^l, ..., z_n^l\}$. The decoder takes atomic representations in and produces features $\{f_1^l, ..., f_n^l\}$ which linearly project to atomic coordinates $\{\hat{c}_1^l, ..., \hat{c}_n^l\}$ (See Figure 2). This might seem like a trivial task, after all the inputs $x_{\text{masked-microenv}}^l$ contain the regression targets. However, since the Graph Transformer only uses relative positions, and only in an attention bias $B^l$, the prediction tasks are quite difficult and require reasoning about the local structure of the microenvironment.

To obtain a residue-level feature representation, we average the atom-level features of the Graph Transformer $z^l = \frac{1}{n} \sum_i z_i^l$ following Diaz et al. (2024). To train this representation, we add $z^l$ into all atomic representations before the decoder. Mathematically, the transformer decoder takes $\{z_1^l + z^l, ..., z_n^l + z^l\}$ as input. We also found that adding this $z^l$ directly to the decoder architecture improves training stability. See Figure 5 for full architecture.

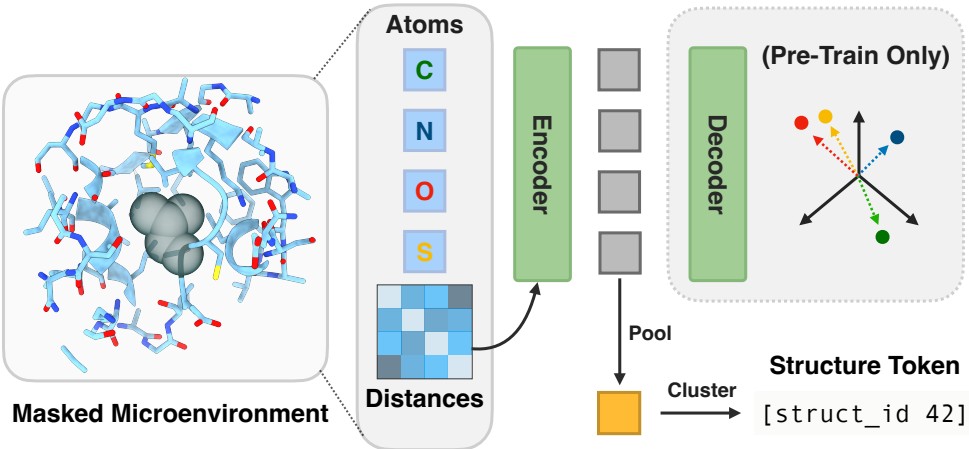

Figure 2: **Atomic Autoencoder learns a structural representation of a residue's microenvironment.** The Autoencoder takes atom element types and pairwise distances as input and reconstructs all atomic coordinates. The encoder is a graph transformer that uses the pairwise distances to bias the attention mechanism to learn rich atomic representations. The atomic representations are pooled to form a microenvironment embedding. The decoder takes the atomic representations and microenvironment embedding as input to decode the coordinates for all atoms. The learned microenvironment embeddings are discretized via K-means into structure tokens, which supervise the fine-tuning of a protein language model. See Figure 5 for architectural details.

**Training objective**. One major challenge is that microenvironments lack robust protein backbone coordinate frames that underpin full protein models (Jumper et al., 2021; Hayes et al., 2024; Dauparas et al., 2022). Unsurprisingly, we empirically observe that vanilla MSE loss $\mathcal{L}_{\text{MSE}}^l = \frac{1}{n} \sum_i \|\hat{c}_i^l - c_i^l\|$ does not take the coordinate frame into account and overestimates the loss. Thus, we optimize the MSE loss after global alignment. First, we employ the Kabsch algorithm (Kabsch, 1976; Umeyama, 1991) to analytically compute the rotation and translation that minimize MSE loss. Then the loss is calculated using the transformed ground truth coordinates. Formally,

$$\mathcal{L}_{\text{MSE-aligned}}^l = \min_{\boldsymbol{R} \in SE(3), \boldsymbol{T} \in \mathbb{R}^3} \frac{1}{n} \sum_i \|\hat{c}_i^l - (\boldsymbol{R} c_i^l + \boldsymbol{T})\|.$$

During training, we observe that naive optimization of the MSE-aligned loss results in convergence to a local optimum where all predicted coordinates lie on a 2-dimensional plane. Following AlphaFold (Jumper et al., 2021), we addressed the issue using a distogram loss. Here, we use ESM3's distogram head by first computing $\boldsymbol{f}_{ij}^l = \boldsymbol{W}_a \boldsymbol{f}_i^l - \boldsymbol{W}_b \boldsymbol{f}_j^l$, where $\boldsymbol{W}_a, \boldsymbol{W}_b$ are linear adapters. We then apply a binned distance loss

$$\mathcal{L}_{\text{disto}}^l = \frac{1}{n^2} \sum_{i,j} \ell_{\text{CE}}(\boldsymbol{C}_{\text{disto}}^T \boldsymbol{f}_{ij}^l, d_{ij}^{\text{bin},l}).$$

where $\boldsymbol{C}_{\text{disto}}$ is a linear classification head that predicts the distance bin $d_{ij}^{\text{bin},l}$ between atoms $i$ and $j$ at residue position $l$. During the first stage of training, we train with the distogram and masked modeling losses, $\mathcal{L}_{disto}^l + \mathcal{L}_{\text{AA}}^l$. During the second stage, we additionally include $\mathcal{L}_{\text{MSE-aligned}}^l$.

**Generating Structure Tokens.** Given a protein structure $\boldsymbol{x}_{\text{struct}}$, we start by generating the masked microenvironment for all $L$ residues, namely $(\boldsymbol{x}_{\text{masked-microenv}}^1, ..., \boldsymbol{x}_{\text{masked-microenv}}^L)$. We feed each masked microenvironment into our Graph Transformer encoder to extract a residue-level feature representation at each position, $(\boldsymbol{z}^1, ..., \boldsymbol{z}^L)$. We quantize $\boldsymbol{z}^l$ for every residue in the protein using K-means (Lloyd, 1982) to generate a structure sequence $\boldsymbol{s} = (s_1, ..., s_L)$. In addition to our autoencoder, we also extract features $(\boldsymbol{z}^{1'}, ..., \boldsymbol{z}^{L'})$ from MutRank (Gong et al., 2024) and generate a second structure sequence $\boldsymbol{s}' = (s_1', ..., s_L')$, both of which are used identically to fine-tune the protein sequence model. Both models are trained on a smaller dataset of experimental structures and are used to generate structure tokens on a large dataset of AlphaFold structures.

Table 1: **Comparisons on structural benchmarks.** We freeze all protein models to assess the learned representation. *ISM* is structure-tuned on the AlphaFold structures of Uniclust30 while *ISM*[†] undergoes additional structure-tuning on PDB structures. SaProt* takes the protein structure as input. All other methods take a sequence as their only input. For contact, secondary structure, and binding residue prediction, the proteins in the training and test sets have at most 30% sequence similarity.

| Method | Structure Prediction (CAMEO) | | | Contact | | | SS | Binding | |
| | GDT-TS | GDT-HA | LDDT | Short | Med | Long | Acc | F1 | MCC |
| --- | --- | --- | --- | --- | --- | --- | --- | --- | --- |
| **Evolutionary pLM** | | | | | | | | | |
| Amplify (Fournier et al., 2024) | - | - | - | 0.38 | 0.36 | 0.23 | 0.82 | 0.22 | 0.26 |
| ESM2 (Lin et al., 2022) | 0.64 | 0.47 | 0.82 | 0.45 | 0.45 | 0.35 | 0.86 | 0.31 | 0.34 |
| ESM2 (fine-tuned) | 0.64 | 0.47 | 0.82 | 0.45 | 0.45 | 0.35 | 0.86 | 0.32 | 0.34 |
| **Structural pLM** | | | | | | | | | |
| ESM2-S (Zhang et al., 2024) | 0.61 | 0.43 | 0.79 | 0.46 | 0.47 | 0.36 | 0.85 | 0.32 | 0.35 |
| S-PLM (Wang et al., 2023) | 0.61 | 0.44 | 0.80 | 0.48 | 0.49 | 0.36 | 0.86 | 0.29 | 0.32 |
| SaProt* (Su et al., 2023) | - | - | - | 0.57 | 0.53 | 0.48 | 0.86 | 0.36 | **0.38** |
| *ISM* (Ours) | **0.67** | **0.50** | 0.83 | 0.61 | **0.60** | **0.49** | **0.89** | 0.35 | 0.37 |
| *ISM* [†] (Ours) | **0.67** | **0.50** | **0.84** | **0.62** | **0.60** | 0.48 | **0.89** | **0.37** | **0.38** |

## 4.2 STRUCTURE-TUNING THE PROTEIN SEQUENCE MODEL

We initialize a sequence-only protein language model trained using masked language modeling (*i.e.,* ESM2) and fine-tune it to predict the structure tokens. We call this training **structure-tuning** and the resulting model *Implicit Structure Model* (*ISM*). We append a linear classification head $C_{\text{struct}}$ to the output of the **pLM** backbone to predict the structural token. The structure prediction loss function is

$$\mathcal{L}_{\text{Struct}} = \frac{1}{|\mathbb{S}|} \sum_{i \in \mathbb{S}} \ell_{\text{CE}}(C_{\text{struct}}^{\top} \mathbf{pLM}(\tilde{\boldsymbol{x}}_{\text{seq}})_i, s_i),$$

where $\tilde{\boldsymbol{x}}_{\text{seq}}$ is the amino acid sequence with masked residues, **pLM** is the protein language model backbone, $\mathbf{pLM}(\tilde{\boldsymbol{x}}_{\text{seq}})_i$ is the representation for residue $i$, $s_i$ is the structure token at residue $i$, and $\mathbb{S}$ are the positions at which the loss is computed. In standard MLM, the loss is computed for all masked positions (*i.e.,* $\mathbb{S} = \mathbb{M}$). We found that predicting structure tokens at *all* positions (*i.e.,* $\mathbb{S} = \{1, ..., L\}$), and not just masked positions, better distills structural representations.

We structure-tune our model on AlphaFold protein structures. AlphaFold sometimes produces inaccurate structures with poorly folded areas showing few interactions. Our structure token visualization reveals that many of these problematic residues are grouped into a single token $s^*$ ([struct id 17] in Figure 3). To ensure data quality, we exclude microenvironments assigned the $s^*$ token from sequence model training. We compute $\mathcal{L}_{\text{Struct}}$ at positions $\mathbb{S} = \{i : i \in \{1, ..., L\}$ and $s_i \neq s^*\}$.

The final training objective for structure-tuning is the sum of structure token(s) and amino acid cross-entropy losses (see Section 3.1), namely $\mathcal{L} = \mathcal{L}_{\text{Struct}} + \mathcal{L}_{\text{MLM}}$.

## 5 RESULTS

### 5.1 IMPLEMENTATION DETAILS

**Atomic Autoencoder.** Our microenvironment-based Atomic Autoencoder is a Graph Transformer encoder with 4 layers and a vanilla Transformer decoder with 2 layers. Our autoencoder training dataset contains 35K proteins from the Protein Data Bank(PDB). We train both stages for 5 epochs with a learning rate of $1 \times 10^{-3}$. See Table 5a for a list of hyperparameters.

**Distillation Dataset.** Once our autoencoder is fully trained, we extract per-residue microenvironment features for 5.8M proteins from Uniclust30 with AlphaFold structures (Mirdita et al., 2017), along with 35K PDB proteins. We identify cluster centroids by applying K-means clustering to features from the PDB database, then assign features to tokens based on their distances to these centroids. The number of clusters, $K = 64$, is chosen using the elbow method. Additionally, we extract per-residue microenvironment features from MutRank and cluster the features into one of $K = 512$ tokens (see Section 3.2).

Table 2: **Comparisons on S669 Single Mutation Thermodynamic Stability prediction.** We compare *ISM* to state-of-the-art methods that take various modalities as input. The middle and bottom block approaches are fine-tuned on cDNA117K, which consists of mini-proteins that have at most 30% sequence similarity with those in S669. UR50: UniRef-50 used in ESM2 pretraining, UR100: UniRef-100, PDB: Protein data bank, UC30: Uniclust30. OAS: Observed Antibody Space. SCOP Structural Classification of Proteins. $r_s$: Spearman correlation coefficient.

| Method | PreTrain Data | $r_s$ | AUC | MCC | RMSE$_\downarrow$ |
|---|---|---|---|---|---|
| FoldX (Schymkowitz et al., 2005) | N/A | 0.27 | 0.62 | 0.14 | 2.35 |
| PROSTATA (Umerenkov et al., 2022) | UR-50 | 0.50 | 0.73 | 0.28 | 1.44 |
| Amplify (Fournier et al., 2024) | UR100,OAS,SCOP | 0.42 | 0.66 | 0.21 | 1.52 |
| S-PLM (Wang et al., 2023) | UR50,SwissProt | 0.41 | 0.68 | 0.18 | 1.53 |
| SaProt (Su et al., 2023) | UR50,UC30 | 0.49 | 0.71 | 0.25 | 1.47 |
| ESM3 (Hayes et al., 2024) | UR70,PDB,MGnify,JGI,OAS,AFDB,ESMAtlas | 0.46 | 0.70 | 0.26 | 1.49 |
| Stability Oracle (Diaz et al., 2024) | PDB | 0.53 | 0.75 | 0.34 | 1.44 |
| MutateEverything (ESM) (Ouyang-Zhang et al., 2024) | UR-50 | 0.47 | 0.72 | 0.31 | 1.48 |
| MutateEverything (AF) (Ouyang-Zhang et al., 2024) | PDB | **0.56** | **0.76** | 0.35 | **1.38** |
| ESM (fine-tuned) | UR-50,PDB+UC30 | 0.49 | 0.72 | 0.25 | 1.47 |
| *ISM* (MutRank only) | UR50,PDB+UC30 | 0.51 | 0.74 | 0.33 | 1.45 |
| *ISM* (MutRank×2) | UR50,PDB+UC30 | 0.50 | 0.73 | 0.32 | 1.45 |
| *ISM* | UR50,UC30 | 0.49 | 0.73 | 0.33 | 1.47 |
| *ISM* | UR50,PDB | 0.52 | 0.74 | 0.30 | 1.45 |
| *ISM* (Ours) | UR50,PDB+UC30 | 0.53 | **0.76** | **0.40** | 1.44 |

**Structure-tuning**. We structure-tune the 650M parameter ESM2 for 20 epochs using a cosine learning rate schedule with 4 warmup epochs. We use a total batch size of 1536 proteins cropped to a maximum sequence length of 512 amino acids. We use AdamW optimizer with a learning rate of $1 \times 10^{-4}$ and weight decay of $5 \times 10^{-3}$. Training takes 26 wall-clock hours on 32 GH200 GPUs. See Table 5b for a complete list of hyperparameters.

## 5.2 COMPARISONS ON STRUCTURE TASKS

Rich sequence representations should inherently capture a protein's fold. In Table 1, we evaluate the structure-enriched representation of *ISM* against established methods on several structure-based downstream tasks, including structure, contact, secondary structure, and binding residue prediction. We evaluate all models as frozen feature extractors and learn a decoding head. For structure prediction, we initialize from pre-trained SoloSeq (Ahdritz et al., 2022), replace the ESM2 backbone model with a frozen protein model, and tune the folding head. For other downstream tasks, we freeze the backbone model and train a shallow head. Contact, secondary structure, and binding residue prediction are evaluated using sequence similarity splits of 30%, 25%, and 20% respectively. More dataset descriptions are listed in Section D. ESM (fine-tuned) follows the same training regimen as *ISM*, but is trained only with masked language modeling. We report results for *ISM* trained on Uniclust30 alone and Uniclust30+PDB.

Our model outperforms all sequence-only models and matches structure-sequence models on all structure-based benchmarks. Notably, on long-range contact prediction, *ISM* outperforms ESM2 by 40%, with a precision of 0.49 against 0.35. This matches the performance of SaProt (0.48), which explicitly requires the structure as input while *ISM* is a sequence-only model. On structure prediction, *ISM* outperforms ESM2 by 5% on the GDT-TS metric (0.67 vs 0.64). On binding residue prediction F1 metric, *ISM* performs similarly with SaProt's 0.36, achieving 0.35 when trained on Uniclust30 and 0.37 when trained on Uniclust30+PDB. Overall, the structure-enriched representations of *ISM* improve performance on various structure-based downstream tasks compared to sequence-only pLMs and structural pLMs.

## 5.3 COMPARISONS ON MUTATION STABILITY EFFECT

Thermodynamic stability is an important phenotype that often needs to be improved during the engineering of a commercially viable protein (Diaz et al., 2023; Liu et al., 2024; Carceller et al., 2024). We evaluate how effectively *ISM* predicts the impact of single mutations on a protein's

Table 3: **System-level Comparisons of transfer learning to various functional benchmarks.** We fine-tune all models with a shallow head for each benchmark (except HumanPPI, in which we freeze *ISM* due to overfitting). * reports the best checkpoint found during training.

| Method | Thermostability | HumanPPI | Metal Bind | EC | GO | | | DeepLoc | |
|---|---|---|---|---|---|---|---|---|---|
| | | | | | MF | BP | CC | Subcell. | Binary |
| | Spearman $\rho$ | Acc | Acc | Fmax | Fmax | Fmax | Fmax | Acc | Acc |
| ESM1b | 0.71 | 0.82 | 0.74 | 0.87 | 0.66 | 0.45 | 0.47 | 0.80 | 0.92 |
| MIF-ST | 0.69 | 0.76 | 0.75 | 0.81 | 0.63 | 0.38 | 0.32 | 0.79 | 0.92 |
| ESM2* | 0.70 | 0.88 | 0.74 | 0.87 | **0.67** | **0.49** | 0.51 | **0.85** | **0.94** |
| SaProt* | **0.72** | 0.88 | **0.79** | **0.88** | 0.65 | **0.49** | 0.51 | **0.85** | 0.93 |
| *ISM* * | 0.71 | **0.89** | 0.75 | **0.88** | **0.67** | 0.47 | **0.52** | 0.84 | 0.93 |

thermodynamic stability ($\Delta\Delta G$) on the S669 dataset (Pancotti et al., 2022) in Table 2. We evaluate all pLMs (ESM, Amplify, S-PLM, *ISM*) identically by fine-tuning the model with a shallow decoder head as in MutateEverything (Ouyang-Zhang et al., 2024). We fine-tune on the cDNA117K dataset from Diaz et al. (2024), a subset of the cDNA display proteolysis dataset (Tsuboyama et al., 2023) where all proteins have at most 30% sequence similarity to those in S669.

*ISM* outperforms all existing models that take a single sequence as input, achieving a Spearman correlation of 0.53 compared to Mutate Everything (ESM)'s 0.49, and an AUC of 0.76 compared to Mutate Everything (ESM)'s 0.72. Additionally, *ISM* matches the performance of state-of-the-art models while only using the amino acid sequence input, achieving an AUC of 0.76, while Mutate Everything (AF) and Stability Oracle achieve AUCs of 0.76 and 0.75, respectively. *ISM* also runs 20× faster on a protein of 300 amino acids. Note that Stability Oracle (Diaz et al., 2024) takes the atomic microenvironment as input and Mutate Everything (AF) (Ouyang-Zhang et al., 2024) takes a multiple sequence alignment as input.

We validate the effectiveness of Atomic Autoencoder by comparing with *ISM* variants structure-tuned with one or two independently trained MutRank models. By incorporating Atomic Autoencoder, *ISM*'s Spearman correlation increases from 0.50 to 0.53 and the AUC increases from 0.73 to 0.76.

We conducted an ablation study on the datasets used for structure-tuning and were surprised to find that training on the smaller PDB dataset enhances downstream $\Delta\Delta G$ performance more than training on the larger Uniclust30 dataset. Specifically, *ISM* achieves a Spearman correlation of 0.49 when trained on UniClust30, compared to 0.52 when trained on PDB. Even though the supervision signal during structure-tuning is derived solely from the atomic coordinates in the structure and not $\Delta\Delta G$ labels, we suspect the PDB dataset has some overlap with the structures in the S669 dataset, resulting in performance similar to that of structure-input models. Overall, on the S669 $\Delta\Delta G$ test set, *ISM* is competitive and even outperforms SOTA structure-based methods and AlphaFold's representations, a feat sequence-only pLMs have yet to achieve.

## 5.4 COMPARISONS ON A DIVERSE SET OF FUNCTIONAL PHENOTYPES

Functional characterization of proteins through biochemical techniques is typically the most resource-intensive type of labeled data to generate, making accurate transfer learning predictions particularly valuable for downstream bioinformatics and protein engineering and design tasks (Yu et al., 2023; Allman et al., 2024; Kulikova et al., 2021). In Table 3, we evaluate *ISM* on the PEER (Xu et al., 2022) and FLIP (Dallago et al., 2021) benchmarks, which encompass tasks that benefit from structural representations (*e.g.,* thermostability), evolutionary representations (*e.g.,* biological process), or both (*e.g.,* EC). We fine-tune all models with a shallow readout head on all benchmarks, except HumanPPI, for which we perform linear probing on *ISM* to prevent overfitting. We observed that longer training leads to overfitting, therefore, we evaluate various training checkpoints and report the highest performance for ESM2, SaProt, and *ISM*. ESM1b (Rives et al., 2019) and MIF-ST (Yang et al., 2023) results are sourced from SaProt (Su et al., 2023).

*ISM* performance remains competitive with ESM2 and other pLMs on functionally diverse tasks and does not stand out. For example, for predicting gene ontology - molecular function (GO-MF), both *ISM* and ESM2 achieve 67% accuracy while SaProt achieves 65%. This finding aligns with

Table 4: **ISM ablation experiments.** Default settings are marked in grey. See Section 6.1. ss: Secondary Structure prediction, mc: MutCompute, mr: MutRank, ae: Atomic Autoencoder

| (a) **Other Structure Tokens** | | | | (b) **Our Structure Tokens** | | | | (c) **Number of clusters** | | | |
|---|---|---|---|---|---|---|---|---|---|---|---|
| tokenizer | contact | ss | bind | tokenizer | contact | ss | bind | $K$ | contact | ss | bind |
| foldseek | 0.42 | 0.88 | 0.32 | ae | 0.38 | 0.88 | 0.35 | 32 | 0.27 | 0.84 | 0.33 |
| esm3 | 0.18 | 0.85 | 0.11 | mr | 0.46 | 0.88 | 0.34 | **64** | **0.48** | **0.89** | **0.37** |
| mc+mr | 0.45 | 0.88 | 0.36 | mr $\times$ 2 | **0.52** | 0.88 | 0.36 | 128 | 0.42 | 0.85 | **0.37** |
| ae+mr | **0.48** | **0.89** | **0.37** | ae+mr | 0.48 | **0.89** | **0.37** | | | | |

| (d) **Pre-training Crop length** | | | (e) **Label Type** | | | (f) **Initialization** | | |
|---|---|---|---|---|---|---|---|---|
| crop | val acc | contact | label | contact | $r_s$ ($\Delta\Delta G$) | init | val acc | contact |
| 32 | 0.27 | 0.27 | features | 0.36 | 0.49 | random | 0.36 | 0.10 |
| 128 | 0.36 | 0.42 | tokens | **0.46** | **0.51** | esm2 | **0.40** | **0.48** |
| 512 | **0.40** | **0.48** | | | | | | |

prior work (Hu et al., 2022), which demonstrates that ESM2 outperforms Evoformer, the feature extractor for AlphaFold, on some functional tasks. It seems that for these functional tasks, the evolutionary signal from masked language modeling is sufficient and does not necessarily benefit from AlphaFold representations. Nonetheless, these experiments demonstrate that the structure-enriched representations of *ISM* do not corrupt ESM2's evolutionary representation on various function-based downstream tasks while enhancing ESM2's structural understanding.

## 6 ANALYSIS

### 6.1 ABLATIONS

We ablate key design decisions by reporting long-range Precision at L (P@L) for contact prediction, accuracy for secondary structure prediction, F1 for binding residue prediction, and Spearman correlation for $\Delta\Delta G$ prediction in Table 4. We also report the validation accuracy, indicating how often the *ISM* variant correctly predicts the structure token derived from Atomic Autoencoder.

**Structure Tokens**. In Table 4a, we distill from various structure models from the literature. We compare against a variant using both MutComputeX-GT (mc) and MutRank (mr) structure models. Since Atomic Autoencoder uses the MLM loss $\mathcal{L}_{AA}^l$ from MutComputeX-GT, this variant determines the effect of dropping the autoencoder from structure-tuning *ISM*. Our model outperforms MutRank and MutComputeX-GT, indicating that the autoencoder provides important structural information.

We found that structure-tuning with ESM3's VQVAE (Hayes et al., 2024) structure tokens do not produce robust structural representations. A model structure-tuned with ESM3 achieves 0.18 and 0.11 on contact and binding residue prediction, compared to 0.48 and 0.37 for *ISM*, respectively. We observe that the accuracy of ESM3 structure token prediction on a held-out validation accuracy on UniClust30 is ~8%, while Atomic Autoencoder accuracy is ~40% and MutRank accuracy is ~47%. We suspect that the large vocabulary of ESM3's VQVAE (4096 structure tokens) results in redundant and overlapping tokens that are difficult to discern and complicate loss optimization.

We also evaluate the performance of our sequence model structure-tuned on FoldSeek VQVAE structure tokens (van Kempen et al., 2022). We train on a larger subset of UniClust30 obtained from SaProt (Su et al., 2023) for the same number of iterations as in *ISM*. The model achieves a long-range contact P@L of 0.42 and a binding residue F1 score of 0.32, which are improvements over ESM3 structure tokens and surpasses the ESM2 baseline (F1 scores of 0.35 and 0.31, respectively). However, representations learned from FoldSeek's VQVAE structure tokens lag behind *ISM* (0.48 and 0.37). Thus, the structure tokens from Atomic Autoencoder and MutRank produce better structure representations, their combination being the most effective (see Table 4b).

**Training parameters**. We evaluate how the maximum length of a sequence during structure-tuning affects the accuracy and downstream performance in Table 4d. When the crop length is dropped to 128 and 32 amino acids, the contact long-range P@L drops from 0.48 to 0.42 and 0.27 respectively. This shows that training with longer sequences is essential for learning long-range contacts.

Additionally, we evaluate the effectiveness of clustering MutRank representations $\boldsymbol{z} = (\boldsymbol{z}^1, ..., \boldsymbol{z}^L) \in \mathbb{R}^{L \times D}$ into tokens $\boldsymbol{s} = (s_1, ..., s_L) \in \{1, ..., K\}^L$ in Table 4e (excluding Atomic Autoencoder supervision). Our model variant uses a linear head to predict the MutRank representations $\boldsymbol{z}$ and is trained with normalized MSE loss. Direct MutRank representation prediction achieves 0.36

P@L, while token ID prediction reaches 0.46 P@L on long-range contact prediction. Clustering the MutRank representations potentially removes superfluous high-frequency noise.

**Evolutionary Pre-Training**. We evaluate the significance of training with MLM before structure tuning in Table 4f by initializing with random weights. This approach resulted in decreased accuracy of structure tokens from 40% to 36%. On downstream contact prediction, training from scratch drops long-range P@L from 0.48 to 0.1. This highlights the importance of structure-tuning a pretrained ESM2 as opposed to structure-tuning from scratch.

## 6.2 QUALITATIVE VISUALIZATIONS

In Figure 3, we visualize atomic structures of microenvironments grouped by structure token id. Specifically, we examine tokens [struct id 3] and [struct id 17], which are the least and most frequently observed tokens in Uniclust30, respectively. We find that microenvironments of the same structure token are semantically related. For example, [struct id 3] contains semi-exposed residues. Interestingly, [struct id 17] includes both solvent-exposed residues from experimental structures and unfolded residues from AlphaFold structures. These findings motivate us to exclude [struct id 17] from our structure-tuning training objective (see Section 4.2). Additional visualizations and analysis are provided in Section E.

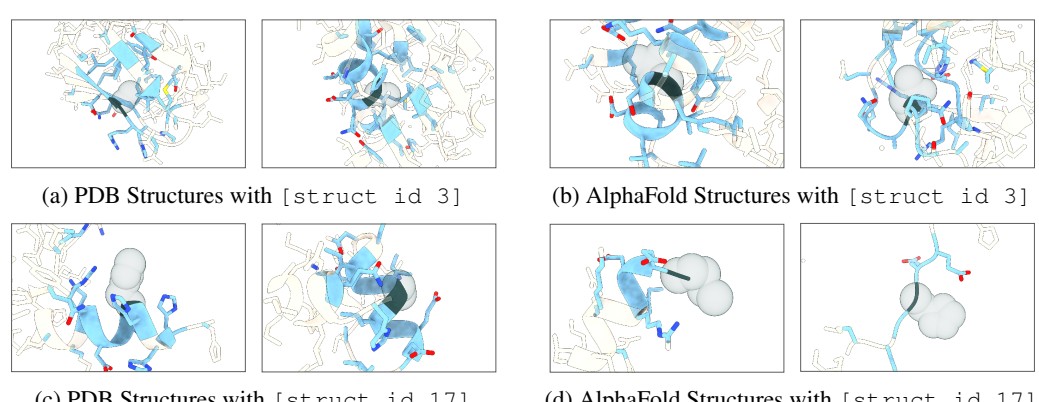

(a) PDB Structures with [struct id 3]   (b) AlphaFold Structures with [struct id 3]

(c) PDB Structures with [struct id 17]   (d) AlphaFold Structures with [struct id 17]

Figure 3: **Cluster-based Microenvironment Visualizations**. Residues in sky blue are within the microenvironment, while white residues are outside and included for context. The grey density indicates the masked-out amino acid. Nodes are colored by element: blue for nitrogen, red for oxygen, and yellow for sulfur. The left two columns display structures from the PDB, while the right two columns show protein sequences from Uniclust30, folded using AlphaFold. [struct id 3] contains semi-solvent exposed residues. [struct id 17] contains solvent exposed residues.

## 6.3 RUNTIME

We compare our runtime against SaProt (Su et al., 2023) on three proteins with 91, 355, and 689 amino acids. The transformer is run on an A40 GPU. Colabfold structure prediction (Mirdita et al., 2022) dominates the runtime. Even with structures, the *ISM* runs 2.4× faster than SaProt which additionally runs FoldSeek (van Kempen et al., 2022) to tokenize the structure.

|  | SaProt | *ISM* (Ours) |
|---|---|---|
| ColabFold | 418 s | - |
| FoldSeek | 43 ms | - |
| Transformer | 28 ms | 28ms |

Figure 4: **Runtime comparison.**

## 7 ACKNOWLEDGEMENTS

This work is supported by the NSF AI Institute for Foundations of Machine Learning (IFML) and UT-Austin Center for Generative AI. We would like to thank AMD for the donation of computational hardware and support resources from its HPC Fund. We acknowledge the Texas Advanced Computing Center at The University of Texas at Austin for providing computational resources (Vista cluster) to support this work.

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
