## A    ATOMIC AUTOENCODER ARCHITECTURE DETAILS

In Figure 5, we visualize the details of our Atomic Autoencoder architecture. We use a GraphTransformer encoder and a vanilla transformer decoder.

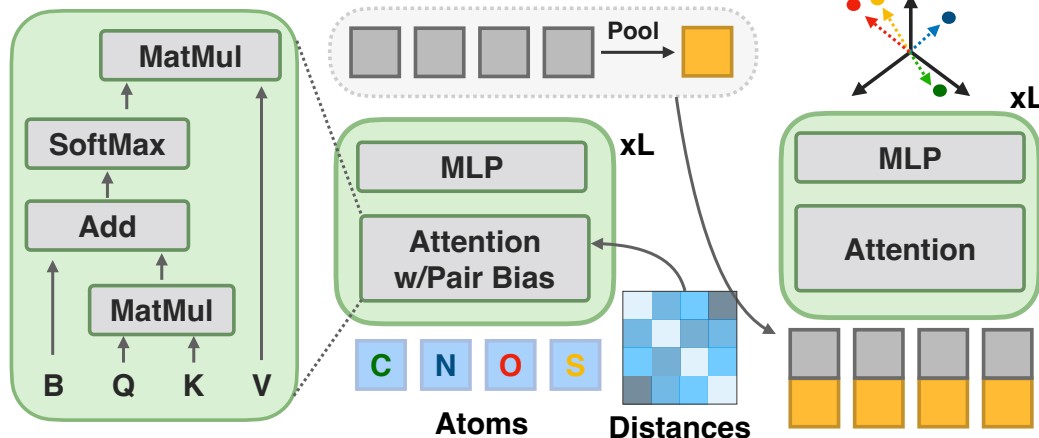

Figure 5: **Atomic Autoencoder Architecture Details.** The autoencoder takes atom element types and pairwise distances as input and reconstructs all atomic coordinates. The encoder is a graph transformer that uses the pairwise distances to bias the attention mechanism to learn rich atomic representations. The atomic representations are pooled to form a microenvironment embedding. The decoder takes the atomic representations and microenvironment embedding as input and produces coordinates for each atom. The learned microenvironment embeddings are discretized via K-means into structure tokens, which supervise the fine-tuning of a protein language model.

## B    ATOMIC AUTOENCODER DATASET

We downloaded a list of proteins from the PDB via PISCES (https://dunbrack.fccc.edu/pisces/) on October 23rd, 2023. We use the 95% sequence similarity split with 37,907 protein chains. We keep all proteins resolved by X-ray crystallography with resolution better than 3Å with no residue breaks and sequence length between 40 and 10,000. After our data pipeline and additional filtering, we ended up with 35,985 proteins in our PDB training set.

## C ATOMIC AUTOENCODER TRAINING AND *ISM* STRUCTURE-TUNING

Table 5 lists the hyperparameters used for training the Atomic Autoencoder (see Section 4.1) and structure-tuning the PLM (see Section 4.2).

Table 5: **Model Hyperparameters.**

(a) **Atomic Autoencoder Training**

| Hyperparameter | Stage 1 | Stage 2 |
|---|---|---|
| *optimization* | | |
| total batch size | 2048 | 2048 |
| optimizer | AdamW | AdamW |
| learning rate | 1e-3 | 1e-3 |
| weight decay | 1e-5 | 1e-5 |
| epochs | 5 | 5 |
| warmup epochs | 1 | 1 |
| clip max norm | 1.0 | 1.0 |
| *modeling* | | |
| layers | 4 | 4 |
| max atoms | 512 | 512 |
| max atom distance | 10.0 | 10.0 |
| *losses* | | |
| $\lambda_{\text{AA}}$ | 1.0 | 1.0 |
| $\lambda_{\text{Distogram}}$ | 1.0 | 1.0 |
| $\lambda_{\text{MSE-aligned}}$ | 0 | 1.0 |
| number of GPUs | 8 | 8 |
| runtime | ~12hr | ~12hr |

(b) **Protein Language Model Structure-tuning**

| Hyperparameter | Structure-tuning |
|---|---|
| *optimization* | |
| total batch size | 1536 |
| optimizer | AdamW |
| learning rate | 1e-4 |
| weight decay | 5e-3 |
| epochs | 20 |
| warmup epochs | 4 |
| clip max norm | 5.0 |
| *modeling* | |
| layers | 33 |
| mask ratio | 15% |
| crop length | 512 |
| *losses* | |
| $\lambda_{\text{MLM}}$ | 1.0 |
| $\lambda_{\text{struct1}}$ | 1.0 |
| $\lambda_{\text{struct2}}$ | 1.0 |
| number of GPUs | 32 |
| runtime | 26hr |

Table 6: **Structural Dataset Statistics.** We report the primary metrics and number of proteins. The split similarity is the maximum allowed sequence similarity between any protein in the training set and any protein in the validation or test sets.

| Dataset | Metrics | Train | Valid | Test | Split Similarity |
|---|---|---|---|---|---|
| Structure Prediction | GDT-TS | 121,481 | - | 185 | - |
| Contact Prediction | Long Range Precision | 25,299 | 224 | 40 | 30% |
| Secondary Structure Prediction | Accuracy | 8,678 | 2170 | 513 | 25% |
| Binding Residue Prediction | F1 | 1,014 | - | 300 | 20% |

Table 7: **Hyperparameters on downstream structural benchmarks.** $^{\star}$: we find that training converges and terminate training early.

| Hyperparameter | Structure | Contact | Secondary Structure | Binding Residues |
|---|---|---|---|---|
| *optimization* | | | | |
| total batch size | 128 | 16 | 16 | 32 |
| optimizer | LION | AdamW | AdamW | AdamW |
| learning rate | 1e-4 | 0.01 | 3e-4 | 1e-4 |
| weight decay | 5e-3 | 0.01 | 0.5 | 0.5 |
| epochs | 20 | 30 | 10 | 10 |
| warmup epochs | 4 | - | 2 | 2 |
| clip max norm | 5.0 | - | 5.0 | 5.0 |
| freeze backbone | True | True | True | True |
| number of GPUs | 32 | 8 | 4 | 8 |
| runtime | 20hr | 40m$^{\star}$ | 35m | 5m |

# D    DOWNSTREAM STRUCTURAL BENCHMARK DETAILS

We summarize our structural datasets in Table 6. In Table 7, we report the hyperparameters used for fine-tuning on different downstream benchmarks. Additionally, we report all additional metrics for contact prediction and binding residue prediction in Table 9 and Table 10 respectively.

## D.1    STRUCTURE PREDICTION

We train on proteins in the PDB and evaluate our model on the CAMEO dataset. Notably, unlike most benchmarks, CAMEO evaluations customarily do not include a sequence similarity split.

We initialize our model from SoloSeq Ahdritz et al. (2022) and freeze our *ISM* backbone. We fine-tune the folding trunk for 10 epochs using a cosine learning rate schedule with 2 warmup epochs. We use a batch size of 128 proteins. We use LION optimizer with a learning rate of $1 \times 10^{-4}$ and weight decay of $0.01$.

We also include comparisons to SoloSeq below. We found that fine-tuning SoloSeq, even with the ESM-2 backbone, improved performance.

Table 8: **System-level Comparisons to prior work on CAMEO structure prediction.**

| Method | GDT-TS | GDT-HA | LDDT |
|---|---|---|---|
| SoloSeq | 0.61 | 0.43 | 0.79 |
| with ESM-2 | 0.64 | 0.47 | 0.82 |
| with *ISM* | 0.67 | 0.50 | 0.83 |

## D.2    CONTACT PREDICTION

We follow the experimental setting as in SaProt (Su et al., 2023), which uses the contact prediction benchmark proposed by Rao et al. (2019) and Xu et al. (2022). In this benchmark, the goal is to predict whether a pair of residues is within a certain distance of one another. We evaluate our model on the ProteinNet CASP12 test set which contains at most 30% sequence identity to those in the training set.

In the main paper, we report precision at L (P@L) for long-range contacts at least 24 amino acids away. In Table 9, we thoroughly evaluate precision at L, L/2, L/5 on short, medium, and long-range intervals of [6,12], [12,24],[24,∞] amino acids respectively. The results of our baseline Amplify model closely align with those reported in their paper.

Table 9: **Comparisons to prior work on contact prediction.** *ISM* is structure-tuned on Uniclust30 while *ISM* [†] is additionally trained on the PDB. SaProt* takes the structure as input. The proteins in the training and test sets have at most 30% sequence similarity.

| Method | Short Range | | | Medium Range | | | Long Range | | |
|---|---|---|---|---|---|---|---|---|---|
| | P@L | P@L/2 | P@L/5 | P@L | P@L/2 | P@L/5 | P@L | P@L/2 | P@L/5 |
| ESM-2 | 0.45 | 0.45 | 0.50 | 0.45 | 0.45 | 0.54 | 0.35 | 0.42 | 0.52 |
| ESM-2S | 0.46 | 0.46 | 0.50 | 0.46 | 0.47 | 0.54 | 0.36 | 0.43 | 0.52 |
| Amplify | 0.38 | 0.38 | 0.41 | 0.36 | 0.35 | 0.40 | 0.23 | 0.28 | 0.35 |
| S-PLM | 0.49 | 0.49 | 0.55 | 0.48 | 0.49 | 0.57 | 0.36 | 0.43 | 0.54 |
| SaProt* | 0.57 | 0.57 | 0.64 | 0.53 | 0.55 | 0.66 | 0.48 | **0.60** | **0.74** |
| *ISM* (Ours) | **0.62** | **0.62** | 0.67 | **0.60** | **0.61** | **0.68** | **0.49** | 0.57 | 0.69 |
| *ISM* [†] (Ours) | **0.62** | **0.62** | **0.68** | **0.60** | 0.60 | **0.68** | 0.48 | 0.56 | 0.67 |

### D.3    Secondary Structure

We use the secondary structure prediction benchmark from Xu et al. (2022). The protein's secondary structures are labeled one of three states - coil, strand, or helix. The training set is taken from Klausen et al. (2019), which contains proteins with no more than 25% sequence similarity. The proteins in the test set have at most 25% sequence similarity to those in the training set. We evaluate the model's classification accuracy.

We freeze *ISM* and train a 2-layer classifier for 10 epochs using a cosine learning rate schedule with 2 warmup epochs. We use a batch size of 32 proteins. We use AdamW optimizer with a learning rate of $1 \times 10^{-4}$ and weight decay of $0.5$.

### D.4    Binding Residues

We use the binding residues benchmark extracted from BioLip (Yang et al., 2012) prepared in the bindEmbed21 method (Littmann et al., 2021). At the time of dataset generation, they found 104,733 structures corresponding to 14,894 sequences in BioLiP. Upon deduplication at 20% sequence similarity, they ended up with 1314 proteins, of which 1014 were used for training and 300 were used for testing. We evaluate on the binary classification of whether a residue is within $< 2.5\text{Å}$ of a metal ion, nucleic acid, or a small ligand (Littmann et al., 2021).

We freeze *ISM* and train a 2-layer classifier for 10 epochs using a cosine learning rate schedule with 2 warmup epochs. We use a batch size of 32 proteins. We use AdamW optimizer with a learning rate of $3 \times 10^{-4}$ and weight decay of $0.5$. Full results with all metrics are available in Table 10.

Table 10: **Comparisons to prior work on binding residue prediction.** *ISM* is structure-tuned on Uniclust30 while *ISM* [†] is additionally trained on the PDB. SaProt* takes the structure as input. The proteins in the training and test sets have at most 20% sequence similarity.

| Method | Test | | | Independent | | |
|---|---|---|---|---|---|---|
| | F1 | MCC | AUC | F1 | MCC | AUC |
| ESM (Lin et al., 2022) | 0.31 | 0.34 | 0.84 | 0.28 | 0.28 | 0.82 |
| ESM-2S | 0.32 | 0.35 | 0.84 | 0.28 | 0.28 | 0.83 |
| Amplify (Fournier et al., 2024) | 0.22 | 0.26 | 0.81 | 0.19 | 0.18 | 0.79 |
| S-PLM (Wang et al., 2023) | 0.35 | 0.36 | 0.83 | 0.35 | 0.33 | 0.82 |
| SaProt* (Su et al., 2023) | 0.36 | **0.38** | **0.87** | **0.35** | **0.33** | **0.87** |
| *ISM* (Ours) | 0.35 | 0.37 | 0.86 | 0.33 | 0.31 | 0.85 |
| *ISM* [†] (Ours) | **0.37** | **0.38** | 0.86 | 0.34 | 0.32 | 0.85 |

# E  QUALITATIVE ANALYSIS ON THE CLUSTERING RESULTS.

We qualitatively evaluate our clusters both on the experimental structures in PDB and the AlphaFold structures in Uniclust30. First, we measured how many unique token IDs occurred in each protein in Figure 6a. Surprisingly, we observed that over 20% of the proteins contained the same token ID (token [17]) for every residue in the sequence. We then measured the number of times each token appeared in the entire Uniclust30 dataset and found that one token appeared over 20% in total (see Figure 6b). This turns out to be token [17] in Figure 7 which contains disordered regions with little or no secondary or tertiary structures. Interestingly, the microenvironments in PDB with token [17] do contain more sparse environments. This motivated us to remove training on the special token $s^* =$ [17].

We also looked at a few tokens in Figure 7 that either occurred the most/least and report our intuition below. Note that while our intuition can offer some rationale about the clusters, the model may capture relevant microenvironment features that are difficult for humans to interpret.

- [id:3]: In PDB proteins, this cluster consists primarily of semi-solvent exposed microenvironments with masked alanines. In Alphafold proteins, the cluster still contains semi-solvent exposed microenvironments but is not as heavily biased towards alanine. This is the least frequently seen structure token in Uniclust30.

- [id:14]: In PDB proteins, this cluster consists primarily of glycine residues that are solvent-exposed and mainly present in highly dynamic loops, often with little local secondary structure. In Alphafold proteins, we observe similar microenvironments, though not as heavily biased towards glycine. This is the second most frequently seen structure token in Uniclust30. It is the most frequently seen token ID in PDB.

- [id:17] In PDB proteins, this cluster consists primarily of residues that are solvent-exposed. However, in Alphafold proteins, this cluster corresponds to poorly folded regions (*e.g.,* N- and C-terminus residues and low pLDDT regions). This is the most frequent structure token in Uniclust30 and the second least frequent structure token in PDB. Because this token accurately captures poorly folded regions in computational structures, we drop this token during training on the Uniclust30 dataset.

- [id:25]: In PDB proteins, this cluster primarily consists of the tertiary interactions centered on disulfide bridges. In Alphafold proteins, this cluster also captures tertiary interactions of small amino acids, primarily glycine. We suspect that since AlphaFold does not explicitly model post-translation modifications, this cluster is not biased towards compact tertiary structures formed by disulfide bridges, as observed in the PDB. This is the least frequently seen structure token in PDB proteins.

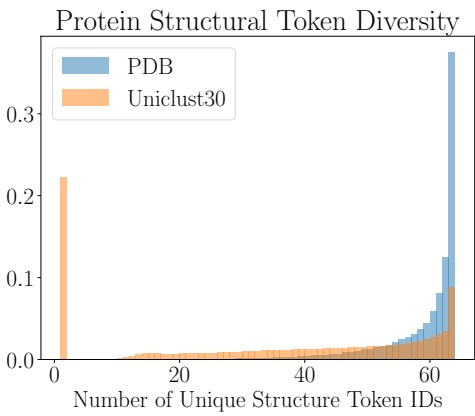

(a) **Histogram of Unique Structure Token IDs per Protein**

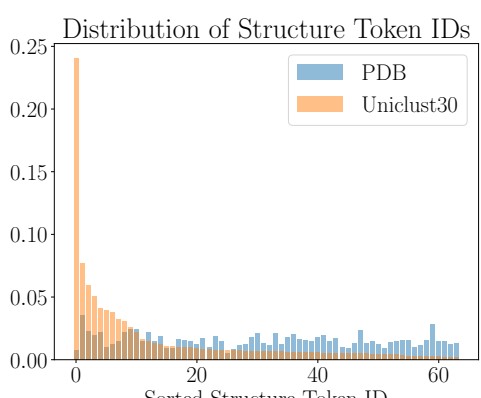

(b) **Distribution of tokens across the entire dataset**

Figure 6: **Measuring the diversity of tokens in both PDB and Uniclust30.**

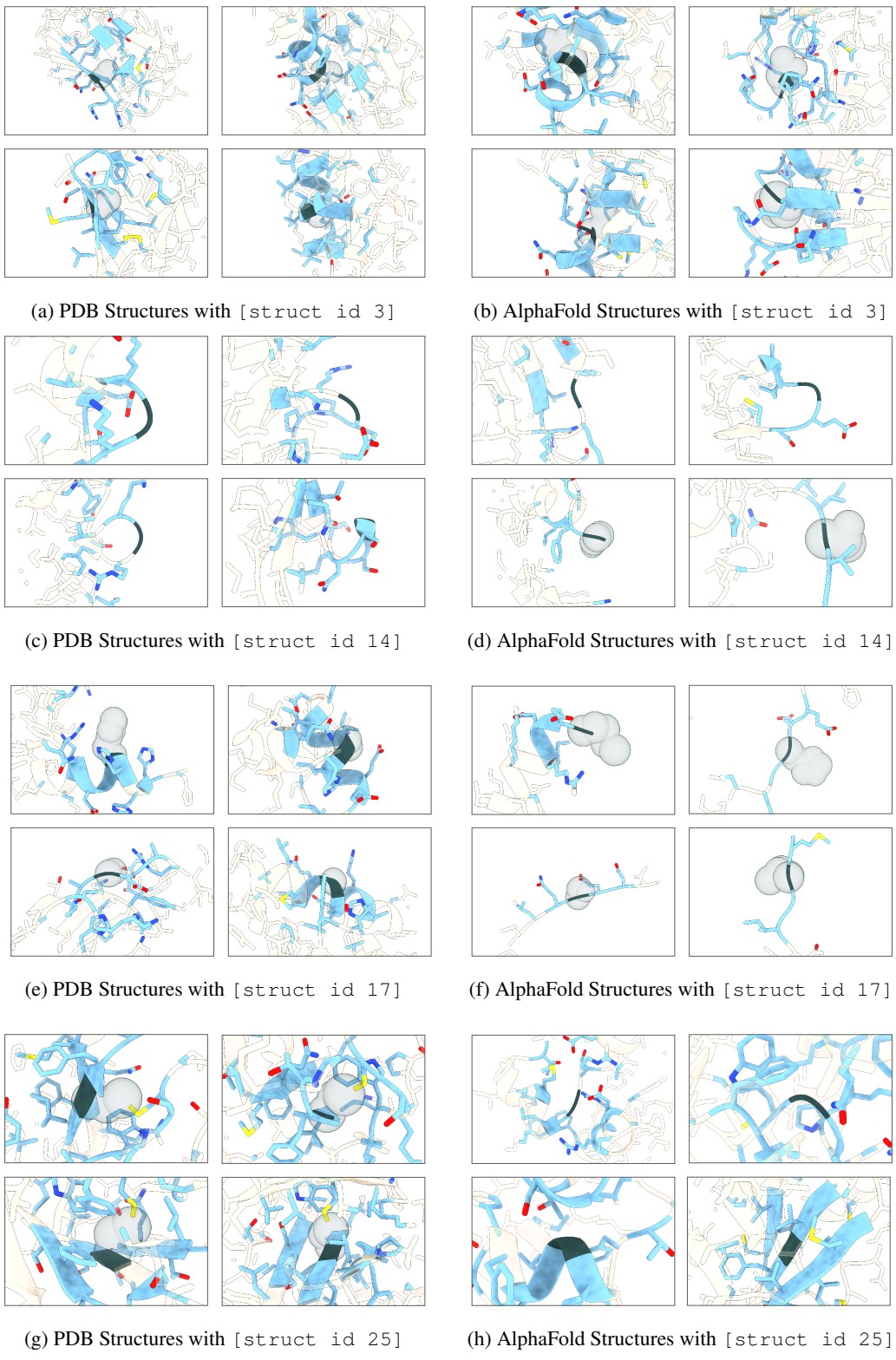

Figure 7: **More Cluster-based Microenvironment Visualizations**. Residues in sky blue are within the microenvironment, while white residues are outside and included for context. The grey density indicates the masked-out amino acid. Nodes are colored by element: blue for nitrogen, red for oxygen, and yellow for sulfur. The left two columns display structures from the PDB, while the right two columns show protein sequences from Uniclust30, folded using AlphaFold.