# OpenReview forum: "Distilling Structural Representations into Protein Sequence Models"
_ICLR.cc/2025/Conference — ICLR 2025 Poster_

### Official Review · Reviewer_eTdZ · 2024-10-20

**Soundness:** 4
**Presentation:** 4
**Contribution:** 2
**Rating:** 6
**Confidence:** 4

**Summary:**

Though AlphaFold2 and its successors have revolutionized protein structure prediction---along with other tasks that require rich embeddings of protein residues---the computational expense of the multiple sequence alignments (MSAs) these models require as input has spurred interest in so-called "single-sequence" models: language models conditioned only on the sequence of the protein in question, and no other structural or co-evolutionary information. So far, these have lagged behind their coevolutionary counterparts on most benchmarks. In an attempt to narrow the gap, the authors propose Implicit Structure Modeling (ISM), which finetunes protein language models with an additional structural loss term for quantized structure tokens. The resulting models can be used as a drop-in replacement for popular single-sequence models but perform better at tasks that require knowledge of structure.

**Strengths:**

The paper is clearly written (with some exceptions---see below), well-motivated, and is fairly comprehensive, including results for most important baselines as well as ablations and analysis. The method is interesting, and it's nice that it's compatible with ESM software.

**Weaknesses:**

While the ISM model compares favorably to others, like SaProt and ESM2, I have two gripes: 1. the difference in performance compared to e.g. SaProt seems fairly marginal across the board, with the strongest improvements in short- and medium-range contact prediction and one of the two thermostability benchmarks (but not the other), and 2.  given the results of the ablations in Table 4, most of the heavy lifting seems to be done by MutRank, not the novel autoencoder. I think I'll need to see a few things before I consider raising my score:

1. Full results of the MutRank model in Tables 1 and 2.
2. Some kind of confidence intervals for the results in Table 4b. It's unclear to me that the small gap there isn't just noise or the result of "ensembling" two different but closely related methods. What if you just trained two independent MR models and ensembled them, for example?
3. More details about how exactly MutRank is incorporated into the model.

There are a few other small details missing as well. Given that the ISM model in Table 1 descends from SoloSeq, you should include SoloSeq results there as well. I think it would also be useful here to include SOTA results (e.g. AlphaFold3), just to illustrate how close ISM is to closing the gap.

Random bits and bobs (no bearing on score):
- The citation for the sentence "MutRank adds a self-supervised..." is wrong.
- There's a typo in Table 1 (Strucure)

**Questions:**

Why do the authors use a custom structure autoencoder and not, say, Evoformer embeddings?
Could you give more details about the PDB dataset used? E.g. what is the cutoff?

---

> ### Author Response · Authors · 2024-11-21
> **Thank you for your diligent review.**
>
> We have incorporated the suggested experiments, baselines and discussion into our revised paper draft (in red).
>
> **Incorporating MutRank in ISM**
>
> During training, ISM incorporates MutRank (MR) using structure-tuning. MutRank takes the atomic microenvironment as input and produces a single microenvironment encoding. The encoding is then clustered into one of K=512 token IDs, which then supervises ISM training. More specifically, we attach a linear MutRank decoding head to ISM and apply cross entropy loss to predict the MutRank token ID. In total, we train ISM with 3 decoding heads to predict the wild-type amino acid, Atomic Autoencoder token ID, and MutRank token ID.
>
> During inference, MutRank is NOT run. Only the ISM sequence model is executed.
>
> **Comparisons with MutRank**
>
> We compare against (1) an ISM variant trained with only MutRank tokens (no Atomic Autoencoder) and (2) another ISM variant trained with two independent MutRank models (ISM MR^2). We obtain the second MutRank token IDs by retraining an independent MutRank.
>
> On the structural benchmarks presented in Table 1, the ISM variant using only MutRank performs comparably to our final ISM model. However, on mutation effect prediction tasks in Table 2, while ISM with only MutRank shows improved performance, the full ISM model, incorporating both MutRank and the Atomic Autoencoder, outperforms it. Critically, only by incorporating the Atomic Autoencoder were we able to achieve performance on par with state-of-the-art methods which use multiple sequence alignments and structures as input.
>
> We observe that ensembling two independent MutRank models does not bridge the performance gap in downstream mutation effect prediction. However, while this approach does not impact performance on structure or secondary structure prediction, it does enhance performance in contact prediction and binding residue tasks.
>
> | Method            | GDT-TS   | GDT-HA   | LDDT     | Short    | Med      | Long     | Acc      | F1       | MCC      |
> | ----------------- | -------- | -------- | -------- | -------- | -------- | -------- | -------- | -------- | -------- |
> | ISM (MR)                  | 0.66     | 0.49   | **0.83** | 0.58 | 0.58 | 0.46 | 0.88     | 0.34 | 0.36 |
> | ISM (MR^2)              | 0.66   | 0.49   | **0.83** | **0.65**  | **0.63** | **0.52** | 0.88 | **0.36** | **0.38** |
> | ISM (MR+AE, Ours) | **0.67** | **0.50** | **0.83** | 0.61     | 0.60     | 0.49 | **0.89** | 0.35 | 0.37 |
>
>
> | Method                 | Spearman | AUC      | MCC      | RMSE ⬇️  |
> | ---------------------- | -------- | -------- | -------- | -------- |
> | Stability Oracle       | 0.53     | 0.75     | 0.34     | 1.44     |
> | Mutate Everything (AF) | **0.56**     | **0.76** | 0.35     | **1.38** |
> | ISM (MR)             | 0.51     | 0.74     | 0.33     | 1.45     |
> | ISM (MR^2)          | 0.50     | 0.73     | 0.32     | 1.45     |
> | ISM (MR+AE) (ours)      | 0.53     | **0.76** | **0.40** | 1.44     |
>
>
> **Comparisons with SaProt**
>
> While the difference in performance compared to SaProt may seem fairly marginal, SaProt explicitly takes the structure as input for downstream tasks. This requires users to have access to the structures and a sophisticated structure-based data pipeline. On the other hand, ISM requires only a sequence as input and is compatible with existing ESM software. We believe this makes ISM a valuable resource for the community despite its similar performance.
>
> We additionally show that our model outperforms SaProt on mutation stability effect prediction with a Spearman correlation of 0.53 vs 0.49.
>
> | Method                 | Spearman | AUC      | MCC      | RMSE ⬇️  |
> | ---------------------- | -------- | -------- | -------- | -------- |
> | SaProt                  | 0.49     | 0.71     | 0.25     | 1.47     |
> | ISM (ours)      | **0.53**     | **0.76** | **0.40** | **1.44**     |

---

> > ### Author Response · Authors · 2024-11-21
> >
> > **Comparisons with SoloSeq/SOTA structure prediction models**
> >
> > We compare to SoloSeq and state-of-the-art structure prediction models. We found that fine-tuning SoloSeq, even with the ESM-2 backbone, improved performance. AlphaFold2 takes a multiple sequence alignment as input and outperforms all single sequence models.
> >
> > | Method     | GDT-TS | GDT-HA | LDDT |
> > | ---------- | ------ | ------ | ---- |
> > | SoloSeq          | 0.61   | 0.43   | 0.79 |
> > | w/ESM2          | 0.64   | 0.47   | 0.82 |
> > | w/ISM              | 0.67   | 0.50   | 0.83 |
> > | AlphaFold2     | 0.75   | -      | 0.89 |
> >
> >
> > **Why use a custom structure autoencoder?**
> >
> > Before developing our custom structure autoencoder, we evaluated several standard options, including FoldSeek VQVAE, ESM3 VQVAE, and Evoformer pair embeddings. However, protein language models structure-tuned on the outputs of these autoencoders did not achieve satisfactory performance (see Table 4a for some quantitative comparisons). This shortcoming motivated us to design a custom structure autoencoder.
> > Our autoencoder improves ISM training because it learns representations for all atoms in a local microenvironment—a level of granularity unexplored in other autoencoders, which primarily focus on the backbone.
> >
> >
> > **PDB Dataset**
> >
> > We downloaded a list of proteins from the PDB via PISCES (https://dunbrack.fccc.edu/pisces/) on October 23rd, 2023. We use the 95% sequence similarity split with 37,907 protein chains. We keep all proteins resolved by X-ray crystallography with resolution better than 3Å with no residue breaks and sequence length between 40 and 10,000.  After our data pipeline and additional filtering, we ended up with 35,985 proteins in our PDB training set.

---

> > > ### Author Response · Authors · 2024-11-22
> > >
> > > Thank you for your time, we hope the reviewer had a chance to look over the rebuttal. Is there anything else the reviewer would need to raise the final rating?

---

> > > > ### Comment · Reviewer_eTdZ · 2024-11-25
> > > >
> > > > Thanks for the hard work. I've raised my score.

---

### Official Review · Reviewer_XC56 · 2024-10-27

**Soundness:** 2
**Presentation:** 2
**Contribution:** 1
**Rating:** 3
**Confidence:** 4

**Summary:**

This paper proposed a protein sequence model enhanced by structure distillation.

**Strengths:**

- The main idea of this paper—integrating protein structure information into a protein sequence model—is clear and direct. However, this concept is not entirely novel and has been explored in previous works, such as “DistilProtBert.”

**Weaknesses:**

- The pretraining task for the Atomic Autoencoder lead to data leakage, raising concerns about its effectiveness.
- Most design choices lack justification. For instance, it is unclear why a two-stage training process was chosen, why multiple loss functions are used, or why ablation studies are absent.
- The performance improvement appears minimal. For example, in comparison to ESM-2, the structure prediction accuracy shows only a slight increase (lddt score: 82 vs. 84).
- The overall contribution of this paper is unclear. The main idea is not novel, the design of the Atomic Autoencoder lacks clarity, and the proposed models offer only marginal performance gains.

**Questions:**

As noted in Weakness 2, most questions arise from the design choices. What is the rationale behind these specific design decisions, and to what extent do they contribute to performance improvement?

---

> ### Public Comment · ~Pranam_Chatterjee1 · 2024-11-15
> **Poorly thought-out review.**
>
> This review poorly summarizes the contributions of ISM and unfairly scores the paper low. While there are "structure-aware" pLMs, DistilProtBert is not one of them—this comparison is not justified. DistilProtBert is simply a distilled version of ProtBert and does not integrate structural tokens or use structural pretraining tasks. In contrast, ISM explicitly integrates structure tokens derived from microenvironments into the sequence model latent space—without requiring the input of actual PDB structures. ISM thus is novel for leveraging structural information implicitly while maintaining the the inherent usability of sequence-only models like ESM-2.
>
> The reviewer's claim that the pretraining task leads to "data leakage" is factually incorrect. The Atomic Autoencoder generates structure tokens by clustering residue embeddings from a structural representation, but the pretraining itself is unsupervised. The structure tokens are derived solely from residues’ local atomic environments. If this constitutes data leakage, then all protein structural models, including AlphaFold and SaProt, would also be guilty of it. Furthermore, ISM operates sequence-only at inference time, meaning there is no reliance on structural inputs beyond the derived tokens used during pretraining.
>
> The reviewer mentions "minimal performance improvement" without acknowledging the typical trend we see in protein modeling: performance improvements on top of high-performing models like ESM-2 are inherently incremental. ISM's improvements, while modest, translate into meaningful downstream benefits in structure-sensitive tasks like binding residue prediction and mutation stability assessment, where ISM provides consistently better results.
>
> The use of multiple loss functions is a feature of ISM. Integrating structure prediction with sequence modeling is a well-established practice in multi-objective optimization of proteins. The manuscript extensively discusses how each component (e.g., structure tokens, structure-tuning loss) contributes to the overall training pipeline. The reviewer’s comment suggests they either overlooked or missed these explanations.
>
> I will say this: ISM is novel in its approach to integrating structure tokens into sequence embeddings without relying on explicit structural inputs during inference. As a user and a developer of pLMs, I can say confidently that this makes ISM significantly more practical than models like SaProt, which require external structural data (e.g., PDB or AlphaFold predictions) at both training and inference stages.
>
> Again, I want to be clear that ISM does introduce a well-motivated, novel strategy for structure-aware protein language modeling. The reviewer’s claims about data leakage and unmotivated design choices are unfounded. The scores provided are wholly unjustified and I encourage the Area Chair to consider this in their final decision.

---

> > ### Author Response · Authors · 2024-11-21
> > **Thank you for the review.**
> >
> > We are glad the reviewer finds the main idea of our paper clear and direct.
> >
> > **Lacks novelty, done in DistilProtBert [1]**
> >
> > Our method and DistilProtBert both use a teacher model to supervise a sequence model using distillation. However, DistilProtBert distills from another sequence model, ProtBert, while ISM distills from a structure model. This cross modality supervision was not studied in DistilProtBert. Additionally, we find that our model outperforms DistilProtBert on secondary structure and binding residue prediction by a large margin.
> > | Method         | Secondary Structure (Acc) | Binding Residue (F1) |
> > |----------------|-------------------------|--------------------|
> > | DistilProtBert | 0.78                    | 0.22              |
> > | ISM                | 0.89                    | 0.35              |
> >
> > [1] DistilProtBert: a distilled protein language model used to distinguish between real proteins and their randomly shuffled counterparts. Geffen et al.
> >
> > **Data Leakage**
> >
> > The Atomic Autoencoder is trained on the PDB. We compare against and match (if not outperform) the performance of structure-based models, such as SaProt and Stability-Oracle. These models take the explicit structures from the PDB as input.
> >
> > **Design choices lack justification (two-stage training, loss functions, ablations missing)**
> >
> > We ablate several decision choices, including distillation losses, initialization, choice of teacher model and more, in Section 6.1 and Table 4. If the reviewer has any particular ablation study they would like to see, we are happy to address this concern.
> >
> > We leverage a two-stage training process with multiple loss functions for Atomic Autoencoder because training with a single autoencoder loss leads to convergence to a local optima where all the atoms lie on the same plane (See Section 4.1).
> >
> > **Performance improvements are minimal compared to ESM2**
> >
> > Let us highlight some of the improvements compared to ESM2.
> > - Precision on long range contact prediction: relative improvement of 40% (0.49 vs 0.35, see Table 1)
> > - F1 score on binding residue prediction: relative improvement of 15% (0.37 vs 0.32, see Table 1)
> > - Spearman correlation for mutation stability effect prediction: relative improvement of 15% (0.53 vs 0.47, see Table 2)
> >
> > **Design of Atomic Autoencoder lacks clarity**
> >
> > We are working on the exposition and are more than happy to address the clarity if the reviewer has any specific questions or comments.
> >
> > We provide a comprehensive overview of Atomic Autoencoder’s design in Figure 2. Section 4.1 offers a detailed discussion of the model’s architecture and loss function. Additionally, a comprehensive illustration of Atomic Autoencoder’s architecture is presented in Appendix Section A for further reference.

---

> > > ### Comment · Reviewer_XC56 · 2024-11-21
> > >
> > > Thank the author very much for the rebuttal. My following comments:
> > > 1.  I apologize for the confusion earlier. I recall coming across similar work before and initially thought it might be related to ‘DistilProtBert.’ However, after further searching, it seems the paper I was referring to is this one (https://arxiv.org/abs/2210.03488). That said, that paper primarily focuses on distilling confidence scores rather than structural information. I’m surprised that this idea hasn’t been explored more extensively before. Perhaps ESM-3 is the closest related work, as it integrates both sequence and structure as input and uses a masked modeling approach to predict one from the other. This seems somewhat analogous to the concept of distillation.
> > >
> > > 2.  The leakage I’m referring to pertains to the pretraining task, not the data split. Specifically, as shown in Figure 2, the pairwise distances are used as input, and the decoder predicts the coordinates of the atoms. This task strikes me as quite problematic for two main reasons:
> > >
> > > 	a.	If the coordinates can indeed be fully reconstructed from pairwise distances, then the task inherently involves information leakage.
> > >
> > > 	b.	If we consider SE(3)-equivariance, using only pairwise distances as input makes this task fundamentally infeasible, as there are infinitely many possible solutions due to arbitrary global rotations.
> > >
> > > I saw you use Kabsch alignment to avoid the infinite solutions, so I guess that pretraining is leakage.

---

> > > > ### Author Response · Authors · 2024-11-21
> > > > **Thank you for the discussion.**
> > > >
> > > > 1. We were also initially surprised that the idea has not been explored more extensively. To clarify, we are not the first to develop a protein sequence model that incorporates enhanced structural representations (see “Sequence Models with Structure Loss” in the Related Works section). However,  distilling structural information into sequence models had previously been challenging due to the lack of an autoencoder capable of capturing the structure with the necessary granularity. Before building Atomic Autoencoder, we experimented with standard approaches, such as FoldSeek and ESM3 VQ-VAEs, as outlined in Table 4a.
> > > > ESM3 takes in multiple input modalities including sequence and structure and is trained with a multimodal masked modeling loss. We find that our model outperforms a sequence-only ESM3 on mutation effect prediction. We have incorporated this comparison into our paper draft.
> > > >
> > > > | Method                 | Spearman | AUC      | MCC      | RMSE ⬇️  |
> > > > | ---------------------- | -------- | -------- | -------- | -------- |
> > > > | ESM3             | 0.46     | 0.70     | 0.26     | 1.49     |
> > > > | ISM (ours)      | **0.53**     | **0.76** | **0.40** | **1.44**     |
> > > >
> > > > 2. It is true that the Atomic Autoencoder predicts the atomic coordinates given the pairwise distances as input. This is a standard use of an autoencoder. In fact, we would have preferred to give the autoencoder the atomic coordinates themselves as input! (for technical reasons, our architecture is not capable of doing this). The point of this project is to let the model see structures during training.
> > > > At a higher level, our goal is not to learn structural representations from sequence alone. Training on structures is therefore not data leakage. Instead, we are trying to come up with a model that does not rely on additional inputs outside of the sequence. This is why we use structure models to improve our training procedure.

---

> > > > > ### Comment · Reviewer_XC56 · 2024-11-22
> > > > >
> > > > > I understand that the Atomic Autoencoder is designed to take structural data as input. However, the key issue lies in the effectiveness of its pretraining task. In models like BERT or GPT, the input is a sequence, and the pretraining task is specifically designed to help the model learn meaningful sequence representations. In contrast, for the pretraining task used in the Atomic Autoencoder, I don’t believe a task that involves information leakage is meaningful. There are prior works that propose well-designed pretraining tasks, such as position denoising in Noisy Nodes, which better facilitate learning robust representations.

---

> > > > > > ### Author Response · Authors · 2024-11-22
> > > > > >
> > > > > > We are trying to understand why you think Atomic Autoencoder has "information leakage". Are there other examples of autoencoders that do not suffer from information leakage? Or, do you feel that any training procedure for a sequence model that involves a structure is data leakage?

---

> ### Comment · Reviewer_XC56 · 2024-11-22
>
> Take GPT as an example: its task is defined as $x_{t+1} = f(x_1, x_2, ..., x_t)$, where the target $x_{t+1}$ is completely absent from the input.
>
> For BERT, the task is $x_k = f(x_1, x_2, ..., x_{k-1}, \text{[MASK]}, x_{k+1}, ...)$, where the target $x_k$ is also unknown in the input. Although BERT typically applies masking to 15% of the tokens, I’ve simplified it to just one token here for clarity.
>
> In contrast, for the "Atomic Autoencoder," the task is $x_{\text{coord}} = f(d_{11}, d_{12}, ...)$, which is a trivial task because the encoder already has access to the full structural information (from pair-wise distance $d$), making it straightforward to recover the coordinates $x_{\text{coord}}$. (As discussed in the above thread)
>
> Furthermore, I don't think the "Atomic Autoencoder" fits the definition of a classical autoencoder. Classical autoencoders typically include an information bottleneck, such as dimensionality reduction, to make the learning process more challenging and meaningful. However, the "Atomic Autoencoder" lacks this feature, functioning more like a straightforward encoder-decoder system.

---

> > ### Author Response · Authors · 2024-11-22
> >
> > Perfect. Let’s clear this up.
> >
> > The information bottleneck in the autoencoder architecture comes in the form of the encoding of the pairwise distances. Our backbone uses the pairwise distances in only a bias for the attention weights and only in the encoder. Empirically, this does not give the model enough information to construct the original positions $x_{coord}$ trivially and will force it to learn a meaningful structure representation as discussed in Section 4.1 lines 231-233. To control for this, we did train a trivial “autoencoder” that sees $x_{coord}$ explicitly in its input. The training curves are at the end of the latest paper draft in Figure 6.
> >
> > As we can see, Atomic Autoencoder trains much slower and needs to extract information about the structure to solve the task while the trivial “autoencoder” does not.
> >
> > Did this clear up the questions enough to warrant a change in the rating?

---

> > > ### Comment · Reviewer_XC56 · 2024-11-23
> > >
> > > Thank you to the authors for providing the additional figure on the training loss. While the result aligns with expectations, I still do not believe this constitutes an information bottleneck. Your current argument is that using pairwise distances only in the attention bias creates an information bottleneck, but I disagree:
> > >
> > > 1. As discussed earlier, pairwise distances can fully recover atomic coordinates. Therefore, using pairwise distances does not introduce an information bottleneck.
> > >
> > > 2. Attention bias is not inherently a weak method for introducing information constraints. For instance, in early language transformers, relative positional encoding was effectively implemented using attention bias, and this approach demonstrated strong performance [1]. Similarly, in molecular modeling, using pairwise distances as attention bias is a widely adopted technique and is sufficient to recover atomic coordinates [2].
> > >
> > > Regarding the loss comparison, it is expected that directly using $x_{\text{coord}}$ as input simplifies the task compared to using pairwise distances. However, this does not prove that current approach is a non-trivial task. It only demonstrates that it is less trivial than directly inputting $x_{\text{coord}}$.
> > >
> > > I would suggest that the authors explore incorporating strategies such as *Noisy Nodes* to further enhance the effectiveness of the pretraining strategy.
> > >
> > >
> > >
> > > [1] Exploring the Limits of Transfer Learning with a Unified Text-to-Text Transformer
> > >
> > > [2] Benchmarking Graphormer on Large-Scale Molecular Modeling Datasets

---

> > > > ### Author Response · Authors · 2024-11-23
> > > >
> > > > We thank the reviewer for their comments but we find their points not supported by evidence. The argument brought forward is not falsifiable by construction, and we will just have to agree to disagree. The evidence brought forward in the paper, our response to the review, and the public comment above support the validity of our approach.

---

> > > > > ### Comment · Reviewer_XC56 · 2024-11-24
> > > > >
> > > > > It took me considerable time to explain to the author that predicting coordinates based on pairwise distances is a trivial pretraining task and therefore does not effectively learn meaningful information. However, the author seems unwilling to acknowledge this. As a result, I feel that further discussion would be unproductive and will maintain my current score.

---

### Official Review · Reviewer_edho · 2024-10-30

**Soundness:** 3
**Presentation:** 3
**Contribution:** 2
**Rating:** 6
**Confidence:** 2

**Summary:**

This paper introduces a novel training strategy for protein language model, which distills structural information into protein sequence models. The main contribution of this paper is the Implicit Strucutre Model (ISM). It utilizes a micorenrironment-based Autoencoder for generating structure tokens and designs a training loss to achieve the self-supervised traning objective of distilling structure tokens. The method demonstrates comparative performence on stractural benchmarks and downstream finetuning tasks.

**Strengths:**

1. The microenvironment-based autoencoder enables the generation of structure tokens with implicit structure information;
2. The authors conducted comprehensive evaluations on various tasks and compared with SOTA methods;
3. The method is compatible with any framework built using ESM2.

**Weaknesses:**

1. The performence improvement presented in Table 2 and 3 seems limited. For instance, it seems that MutateEverything (AF) demonstrates better results than ISM even with fewer pretrain data (PDB only); moreover, in table 3, ISM does not demonstrate superior performance than ESM2 and SaProt.
2. The poteintial impact on the community of protein studies is not clearly presented. The authors could better articulate the potential impact, such as providing examples of how ISM could enable new applications or analyses in protein research that were not previously possible with existing models.

**Questions:**

Protein LMs such as AlphaFold and ESMFold take protein sequences as inputs and have the ability to generate structure information. It may indicate that these models also captures structure information implicitly without introducing an "ISM like" module and finetuning strategy. Thus, it makes me confused on the contribution of the method for the community of computational biology.

The authors may directly compare ISM's performance to AlphaFold and ESMFold on structure-related tasks, explain how ISM's approach differs from or improves upon the implicit structure learning in those models.

It would be benifit if the authors could clarify what unique capabilities or advantages ISM provides compared to existing protein language models.

---

> ### Author Response · Authors · 2024-11-21
> **Thank you for your detailed review.**
>
> **Performance improvements are limited: ISM vs. MutateEverything (AF)**
>
> Performance comparisons between ISM and MutateEverything (AF) are nuanced: ISM underperforms MutateEverything (AF) on Spearman correlation (0.53 vs. 0.56) and RMSE (1.44 vs. 1.38) but outperforms on MCC (0.40 vs. 0.35).
>
> Importantly, MutateEverything (AF) relies on a multiple sequence alignment (MSA) as input, which encodes explicit evolutionary information but is computationally expensive to generate and may be unavailable for certain proteins, such as de-novo designs. In contrast, ISM requires only the target sequence as input, offering greater accessibility and versatility. For example, on a 317 amino acid protein, ISM is 20x faster than the AlphaFold-based model (0.6 vs 12.1 seconds).
>
> Note that while AlphaFold was trained on fewer protein sequences in PDB compared to UniRef, it is trained on much richer supervision in the form of experimental structures.
>
> **Performance improvements are limited: Functional Tasks in Table 3**
>
> Our experiments demonstrate ISM's versatility and effectiveness across different tasks. On structural benchmarks and mutation effect prediction (Tables 1 and 2), ISM achieves superior performance compared to existing models. For other benchmarks where structural representations may be less critical (Table 3), ISM performs competitively with state-of-the-art models. These comprehensive results establish ISM as a robust choice for a wide range of protein modeling tasks, even ones requiring less structural knowledge.
>
> **Unique capabilities or advantages ISM provides compared to existing pLMs.**
> ISM’s unique capability lies in its structural-enriched representations from a single sequence input. It does so without computationally or experimentally generating a structure while matching performance of structure-based models on downstream tasks.
>
> **The potential impact on the community of protein studies is not clearly presented.**
>
> Integrating structural information into protein language models (pLMs) while maintaining sequence-only input requirements represents a significant advancement for computational biology. We demonstrate the practical impact of this approach on two structure-dependent applications: (1) improving ESMFold's structure prediction capabilities, and (2) enhancing mutation effect prediction for ΔΔG. This advancement is particularly valuable because many computational biology applications rely on readily available sequence data while being implicitly influenced by unavailable protein structures. ISM's ability to capture structural information from sequence alone makes it applicable to these scenarios.
>
> **Discussion and Comparisons with AlphaFold as structure learner.**
>
> While AlphaFold requires both protein sequence and multiple sequence alignment inputs, ISM learns structural representations directly from sequence alone. This fundamental difference makes AlphaFold an unequal comparison and not the most relevant baseline. Nevertheless, for ΔΔG mutation effect prediction in Table 2, ISM matches the performance of state-of-the-art models, including MutateEverything (AF), which leverages AlphaFold as the backbone.
>
> **Discussion and Comparisons with ESMFold as structure learner.**
>
> ESMFold is a structure predictor which takes a protein sequence as input, derives a latent representation from a frozen ESM-2, then trains a folding trunk on top of the ESM2 representations. We directly compare against an ESMFold variant using the 650M parameters ESM-2 model in Table 1 and find that our model outperforms ESMFold with GDT-TS of 0.67 vs 0.64. Additionally, we compare ISM against ESM-2 representations for all structural benchmarks and found consistent performance improvements in Table 1.

---

> > ### Author Response · Authors · 2024-11-22
> >
> > Thank you for your time, we hope the reviewer had a chance to look over the rebuttal. Is there anything else the reviewer would need to raise the final rating?

---

> > > ### Comment · Reviewer_edho · 2024-11-25
> > >
> > > Thanks for your clarification. I have read your response and other public comments. I decide to keep the score.

---

### Meta-Review · Area_Chair_QatU · 2024-12-19

**Metareview:**

The key innovation in this paper is the Atomic Autoencoder.

Two of the reviewers appreciate the contribution and are in favor of acceptance (score 6) and one is strong against. The authors and reviewers engaged in a good clarifying discussion that however ended up in this mixed outcome.

The key contention is whether the Atomic Autoencoder task is meaningful or not. It is introduced like this in the paper:

"The encoder takes the masked microenvironment x l microenv as input and produces atomic representations {z l 1 . . . z l n}. The decoder takes atomic representations in and produces features {f l 1 . . . f l n} which linearly project to atomic coordinates {ci : ∀(ei , ci) ∈ x l microenv} (See Figure 2). This might seem like a trivial task, after all the inputs x l microenv contain the regression targets. However, since the Graph Transformer only uses relative positions, and only in an attention bias Bl , the prediction tasks are quite difficult and require reasoning about the local structure of the micro-environment."

The empirical results seems to indicate that this is the case.  These show improvements over other models that are not using multiple sequence alignment.

The idea is clear and despite the discussed limitation seems to work in practice. The authors have done very thorough benchmarking and ablation.

**Additional Comments On Reviewer Discussion:**

None.

---

### Decision · Program_Chairs · 2025-01-22

Accept (Poster)